# Calreticulin Regulates SARS-CoV-2 Spike Protein Turnover and Modulates SARS-CoV-2 Infectivity

**DOI:** 10.3390/cells12232694

**Published:** 2023-11-23

**Authors:** Nader Rahimi, Mitchell R. White, Razie Amraei, Saran Lotfollahzadeh, Chaoshuang Xia, Marek Michalak, Catherine E. Costello, Elke Mühlberger

**Affiliations:** 1Department of Pathology, School of Medicine, Boston University, Boston, MA 02118, USA; ramraei@bu.edu; 2Department of Microbiology, School of Medicine, Boston University, Boston, MA 02118, USA; mitchw@bu.edu (M.R.W.); muehlber@bu.edu (E.M.); 3National Emerging Infectious Diseases Laboratories (NEIDL), Boston University, Boston, MA 02118, USA; 4Renal Section, Department of Medicine, Medical Center, Boston University, Boston, MA 02118, USA; slotfoll@bu.edu; 5Center for Biomedical Mass Spectrometry, School of Medicine, Boston University, Boston, MA 02118, USA; csxia@bu.edu (C.X.); cecmsms@bu.edu (C.E.C.); 6Department of Biochemistry, University of Alberta, Edmonton, AB T6G 2H7, Canada; mmichala@ualberta.ca

**Keywords:** COVID-19, SARS-CoV-2, spike protein, S-RBD, calreticulin, intracellular calcium homeostasis, endothelial cells

## Abstract

Cardiovascular complications are major clinical hallmarks of acute and post-acute coronavirus disease 2019 (COVID-19). However, the mechanistic details of SARS-CoV-2 infectivity of endothelial cells remain largely unknown. Here, we demonstrate that the receptor binding domain (RBD) of the SARS-CoV-2 spike (S) protein shares a similarity with the proline-rich binding ena/VASP homology (EVH1) domain and identified the endoplasmic reticulum (ER) resident calreticulin (CALR) as an S-RBD interacting protein. Our biochemical analysis showed that CALR, via its proline-rich (P) domain, interacts with S-RBD and modulates proteostasis of the S protein. Treatment of cells with the proteasomal inhibitor bortezomib increased the expression of the S protein independent of CALR, whereas the lysosomal/autophagy inhibitor bafilomycin 1A, which interferes with the acidification of lysosome, selectively augmented the S protein levels in a CALR-dependent manner. More importantly, the shRNA-mediated knockdown of CALR increased SARS-CoV-2 infection and impaired calcium homeostasis of human endothelial cells. This study provides new insight into the infectivity of SARS-CoV-2, calcium hemostasis, and the role of CALR in the ER-lysosome-dependent proteolysis of the spike protein, which could be associated with cardiovascular complications in COVID-19 patients.

## 1. Introduction

Severe acute respiratory syndrome coronavirus 2 (SARS-CoV-2) is the causative agent of coronavirus disease 2019 (COVID-19). SARS-CoV-2 virions attach to the host cell membranes through the spike glycoprotein (S) by binding to receptors and attachment factors, thereby facilitating viral entry into target cells [1,2,3]. SARS-CoV-2 infects endothelial cells through interactions with CD209L/L-SIGN [1] and vimentin [2]. Although ACE2 is the most common entry receptor for SARS-CoV-2, its expression in endothelial cells is very low or undetectable [1,4]. SARS-CoV-2 S protein’s interaction with the host receptors and attachment factors is largely established through its receptor binding domain (RBD), which is composed of five-stranded anti-parallel β sheets (β1, β2, β3, β4, and β7) with short connecting helices and loops that form the core [5]. However, S-RBD can employ a variety of mechanisms to interact with the host entry receptors and attachment factors. For example, the motif involved in the recognition of ACE2 by the S-RBD is different from the motif involved in the recognition of the attachment factor vimentin [2]. Moreover, the S-RBD is subjected to various posttranslational modifications (PTMs) such as glycosylation and proteolytic modification, which could play a role in the interactions of the S protein with the host proteins [6,7,8].

Recent findings indicate that the SARS-CoV-2 S protein has additional functions beyond its canonical role in entry receptor recognition. For example, when the S protein alone is ectopically expressed in human cells, it induces pro-inflammatory responses in the host cells [9,10,11,12], causes endothelial cell damage, and disrupts the blood–brain barrier function [13], indicating that aside from its well-known canonical role in viral entry, SARS-CoV-2-S protein could play a direct role in the pathogenesis of COVID-19. However, the mechanisms of the S protein in the pathogenesis of COVID-19 remain largely unknown.

As a typical class I fusion protein, SARS-CoV-2 S protein is co-translationally translocated into the endoplasmic reticulum (ER) and undergoes extensive *O*- and *N*-glycosylation [8], participating in virus budding [14] and virus particle assembly [15,16]. In this study, we demonstrate that S-RBD shows certain similarities to drosophila enabled/vasodilator-stimulated phosphoprotein homology 1 (EVH1) domain and interacts with the proline-rich domain of CALR. CALR regulates the S protein turnover and SARS-CoV-2 infectivity of human endothelial cells. This study provides new insight into the critical role of CALR in the regulation of S protein proteostasis and SARS-CoV-2-induced impairment of intracellular calcium homeostasis. The latter action could be associated with the cardiovascular complications of COVID-19.

## 2. Materials and Methods

**Antibodies, Plasmids, and shRNAs.** CALR mCherry plasmid (#55006), YC-Nano15 plasmid (#51962) and full-length SARS-CoV-2 spike plasmid with C-terminus C9 tag (#145031) were purchased from Addgene (Watertown, MA, USA). Human CALR-shRNA (#sc-29234-SH) and anti-CALR antibody (#sc-166837) were purchased from Santa Cruz Biotechnology (Dallas, TX, USA). Anti-spike (RBD) antibody (#703973, 1:200 dilution used in the immunoblotting experiments) was purchased from Invitrogen (Waltham, MA, USA). Anti-mCherry antibody (#43590S, 1:1000 dilution), anti-GAPDH antibody (1:1000), and c-Myc antibody (1:1000) were purchased from Cell Signaling Technology (Danvers, MA, USA). Production of rabbit anti-VEGFR-2 antibody was as previously described [17,18] and used in 1:1000 dilution for the immunoblotting experiments. The CALR-shRNA plasmids (#sc-29234-SH) used in this study were a pool of three to five lentiviral vector plasmids, each of which encoded a target-specific 19 to 25 nt shRNA with a 6-bp loop. SARS-CoV-2 S-RBD-HIS (GenBank: MN975262.1) was cloned into a pVRC vector containing a HRV 3C-cleavable C-terminal SBP-His8x tag) [2]. Construction and expression of GST-fusion CALR constructs were previously described [19].

**Cell culture.** Pathogen-free HEK-293 and HUVEC-TERT cells were maintained in Dulbecco’s modified eagle medium (DMEM) was supplemented with 10% fetal bovine serum (FBS), L-glutamine (2 mM), penicillin (50 units/mL), and streptomycin (50 mg/mL). HEK-293 and HUVEC cells were purchased from ATCC (Manassas, VA, USA). HUVEC cells were transformed with TERT as previously described (47). Pulmonary microvascular endothelial cells (#C-12281) were purchased from PromoCell (Heidelberg, Germany) and maintained in a special endothelial cell media.

**3D modeling of S-RBD and CALR.** The structure of the human EVH1 domain of VASP (1EGX) and SARS-CoV2 S-RBD (7bwj) were downloaded from the publicly available PDB site. The visualization and alignment of the structure of S-RBD with the EVH1 domain were determined via PyMOL software (version 4.6.0).

**Propagation of SARS-CoV-2-mNG and infection of endothelial cells.** SARS Co-V-2 isolate USA_WA1/2020 was kindly provided by Natalie Thornburg and the World Reference Center for Emerging Viruses and Arboviruses (WRCEVA). Recombinant SARS-CoV-2 expressing mNeonGreen (SARS-CoV-2-mNG) was kindly provided by Pei-Yong Shi, from the University of Texas Medical Branch, Galveston, TX, and WRCEVA, which was based on SARS-CoV-2 isolate USA_WA1/2020 [20,21]. SARS-CoV-2-mNG and SARS-CoV-2 stocks were grown in Vero E6 cells and virus titers were determined by tissue culture infectious dose 50 (TCID50) assay, as described previously [20,21]. Work with SARS-CoV-2-mNG and SARS-CoV-2 were carried out in the biosafety level (BSL)-4 facility of the National Emerging Infectious Diseases Laboratories at Boston University following approved standard operating procedures. Infection of cells was carried out as previously described [1,2]. Briefly, cells (5 × 10^4^ per well) were seeded in 96-well plates. The following day, cells were infected with SARS-CoV-2-mNG at a multiplicity of infection (MOI) of 0.2 or 2 (triplicates/condition) or indicated in the figure legends. After 24 h or as indicated in the figure legends, the cells were fixed in 10% PFA for 6 h at 4 °C before being removed from the BSL-4 laboratory. Cells were washed with PBS prior to imaging. Images of cells were acquired using a Nikon deconvolution microscope equipped with a camera (Nikon Corporation, Tokyo, Japan). Quantification of infected cells was carried out via Image J software (version v1.54g).

**Immunoprecipitation and Western blot analysis.** Cells were grown in 10 cm culture dishes in DMEM plus 10% FBS. Cells were lysed and normalized whole cell lysates were subjected to immunoprecipitation by incubation with appropriate antibodies, as described in the figure legends. Immunocomplexes were purified by incubation with protein A-Sepharose/protein G-agarose beads. The immunoprecipitated proteins were subjected to Western blot analysis. In some occasions, membranes were stripped by incubating in a stripping buffer (6.25 mM Tris-HCl, pH 6.8, 2% SDS, and 100 mM β-mercaptoethanol) at 50 °C for 30 min, washed in western rinse buffer (20 mM Tris and 150 mM NaCl), and re-probed with the desired antibody. Quantification of blots was carried out via Image J software (version v1.54g).

**Purification of GST-CALR proteins and GST pull-down assay.** Recombinant GST-CALR proteins were purified from BL21(DE3) *Escherichia coli* transformed with GST-pGEX constructs. The protein expression was induced by 0.1 mM isopropyl-β-D-thiogalactoside (IPTG) at 30 °C overnight. After removing cell debris, GST-fusion proteins were purified via glutathione Sepharose beads. In vitro GST fusion CALR experiments were performed as previously described [2,22].

**Statistical analyses.** Experimental data were subjected to the student *t* test, with representation from at least three independent experiments. *p* < 0.05 was considered significant.

## 3. Results

### 3.1. S-RBD Partially Resembles the EVH1 Domain and Interacts with the Proline-Rich Domain of CALR

Our initial amino acid sequence alignment of S-RBD with the drosophila-enabled/vasodilator-stimulated phosphoprotein homology 1 (EVH1) domain revealed a certain sequence homology between the S-RBD and EVH1 domain (Figure 1A). Although the amino acid sequence alignment of S-RBD with the EVH1 domain showed nearly 9% sequence identity (Figure 1A), the key features of the EVH1 domain interaction with the proline-rich sequences (PRS) [23], which is the presence of specific aromatic residues (e.g., W, F, and Y) on the EVH1 domain that form bonds between the PRS sheets [24,25], were conserved in the S-RBD (Figure 1A, shown in red). The EVH1 domain is also present on actin-based structures such as VASP, EVL, MENA, and HOMER, and is known to mediate protein–protein interaction via recognition of the PRS [24]. The EVH1 domain is composed of seven anti-parallel β sheets that are closed by an α-helix, running approximately parallel to the barrel axis such that it connects strands β1 and β5 (Figure 1B) [24]. Similar to the EVH1 domain, the S-RBD is also composed of five anti-parallel β sheets (β1, β2, β3, β4, and β5) with short connecting helices and loops that form the core (Figure 1B) [5]. However, despite having similar folding pattern, the root mean square deviation (RMSD) value of the alignment was 7.2, indicating that the S-RBD and EVH1 domains do not fully fold in an identical manner.

We recently used a recombinant S-RBD protein as bait and whole-cell lysate of human umbilical vein endothelial cells (HUVEC-TERT) as a source for prey proteins followed by liquid chromatography–tandem mass spectrometry (LC-MS/MS) analysis and identified vimentin as an attachment factor for SARS-CoV-2 and showed that it facilitates viral entry into HUVEC-TERT cells [2]. Multiple other host proteins were also identified as S-RBD binding partners in our LC-MS/MS analysis; one of the prominent proteins was calreticulin (CALR). The spectra of CALR is shown (Appendix A). CALR is composed of three major domains: an N-terminal lectin-like globular domain that is involved in carbohydrate binding; a central PRS that forms an extended arm structure and has high affinity for calcium binding and an acidic calcium binding C-terminal domain [26,27], which functions as a calcium binding/storage protein in the ER [26]. The N-terminal domain together with the P-domain and part of the C-terminal domain are responsible for the chaperone function of CALR.

To corroborate our LC-MS/MS data, we generated recombinant GST-fusion CALR and S-RBD-STRP-HIS proteins (Figure 1C) and tested for the ability of S-RBD-STRP-HIS to interact with the GST-CALR in a modified GST pull-down assay. Expression and preparation of the purified S-RBD were as previously described [1,2]. The results showed that S-RBD-STRP-HIS binds to full-length (FL) CALR in a concentration-dependent manner (Figure 1D). At high concentration (5 µg), S-RBD-STRP-HIS interaction with CALR was readily detected (Figure 1D).

Next, we investigated whether S-RBD interacted with CALR in the cellular context. We co-expressed mCherry-CALR with S-RBD-Myc or with S1-Myc in HEK-293 cells (Figure 2A) and examined the interaction of S-RBD-Myc and S1-Myc with mCherry-CALR via a co-immunoprecipitation assay. The results showed that both S-RBD-Myc and S1-Myc interacted with mCherry-CALR (Figure 2B).

CALR is an ER resident protein. However, it has also been reported that CALR can be expressed extracellularly in certain cells, particularly in cells undergoing apoptosis [28]. Additionally, we analyzed whether the interaction of S-RBD with the CALR is mediated via the P-domain of CALR. We generated a panel of GST-fusion CALR constructs encompassing the N-terminal alone, the N-terminal plus P-domain, the P-domain, and the C-terminal domain alone (Figure 2C). A GST pull-down assay showed that the N-terminal plus P-domain and P-domain alone, but not the N-terminal domain alone or C-terminal domain alone, interacted with the S-RBD (Figure 2D). Next, we generated a P-domain-truncated GST-CALR (GST-ΔP-CALR) construct (Figure 2F) and tested its ability to interact with the S-RBD-Myc. The results showed that the full-length CALR but not the GST-ΔP-CALR interacted with the S-RBD-Myc (Figure 2F). Taken together, our data demonstrate that the P-domain mediates the interaction of CALR with the S-RBD.

### 3.2. Calreticulin Regulates the Turnover of SARS-CoV-2 Spike Protein

We then investigated the functional importance of the interaction of spike protein with CALR. We knocked down CALR via shRNA in HEK-293 cells (Figure 3A) and examined its effect on spike protein turnover. The turnover of the spike protein was determined with a puromycin pulse-chase experiment, where the cells were treated with puromycin (to inhibit the synthesis of new protein). The results showed that the turnover of spike protein after 60 and 120 min puromycin treatment in control HEK-293 cells was 69% and 39%, respectively, whereas the turnover of spike protein in HEK-293 cells in CALR knockdown cells under a similar condition was 100% and 47%, respectively (Figure 3A). To demonstrate whether depletion of CALR is directly responsible for the change in the turnover spike protein, we overexpressed mCherry-CALR in HEK-293 cells expressing CALR shRNA and spike protein. Overexpression of mCherry-CALR rescued the effect of CALR-shRNA and even further reduced the turnover of spike protein. The turnover of spike protein after 60 min puromycin treatment in CALR shRNA/HEK-293 cells was 94% compared to 19% in HEK-293 cells overexpressing mCherry-CALR (Figure 3B). The data suggest that CALR modulates the turnover of spike protein in HEK-293 cells.

### 3.3. Calreticulin Regulates Spike Protein Levels via Lysosomal-Dependent Degradation

Given the profound negative regulatory effect of CALR on spike protein levels, we questioned whether the interaction of CALR with the spike protein could route the spike protein for lysosomal or proteasomal degradation. To address this question, we treated HEK-293 cells expressing spike with control shRNA or co-expressing spike with CALR shRNA with bafilomycin, a potent lysosome/autophagy inhibitor, which functions by inhibiting the acidification of the lysosome [29] and bortezomib, a selective proteasome inhibitor. The results showed that treatment of cells with bortezomib increased the abundance of the spike protein in HEK-293 cells expressing control shRNA or CALR-shRNA (Figure 3C). Overall, bortezomib increased the spike protein levels in spike/HEK-293 and spike/CALR shRNA/HEK-293 cells by 11.4-fold and 13.8-fold, respectively (Figure 3C). This suggests that spike protein is regulated by proteosome-dependent degradation; however, its proteasomal degradation is not significantly influenced by the expression of CALR. Similar to the effects of bortezomib, treatment of HEK-293 cells expressing spike with bafilomycin 1A also increased spike protein levels (Figure 3C), indicating that the spike protein is subject to both lysosome- and proteasome-dependent degradation. However, bafilomycin 1A treatment of HEK-293 cells expressing spike/CALR-shRNA had a greater effect on the spike protein (Figure 3C); bafilomycin 1A at 100 µM increased spike protein levels by 3.24-fold in spike/HEK-293 cells, whereas at the same concentration, it increased the spike protein levels by 7.88-fold in CALR shRNA/HEK-293 cells (Figure 3C), suggesting that CALR protects spike protein from the lysosome-mediated degradation. Curiously, bafilomycin 1A increased the abundance of both the full-length and S1 subunit of the spike protein levels (Figure 3C), indicating that both the full-length (FL) and S1 subunits of the spike protein are subject to lysosome-mediated degradation. To address whether the protective effect of CALR in lysosome-mediated degradation of the spike protein is specific to the spike protein, we examined the effect of knockdown of CALR on VEGFR-2 levels, which undergoes 26S-proteasome-dependent degradation [30,31]. The results showed that knockdown of CALR or bafilomycin 1A treatment have no measurable effect on VEGFR-2 levels (Figure 3D), indicating that the effect of knockdown of CALR on spike protein levels is not a general response of cells to depletion of CALR.

### 3.4. Calreticulin Regulates Infectivity and SARS-CoV-2-Induced Intracellular Calcium Homeostasis in Human Endothelial Cells

Next, we investigated whether CALR plays a role in the infectivity of SARS-CoV-2. We knocked down CALR in transformed human umbilical vein endothelial (HUVEC-TERT) cells, which we have previously shown are susceptible to SARS-CoV-2 infection [1,2] and human primary pulmonary microvascular endothelial cells (PMVEC) (Figure 4A). SARS-CoV-2 infection of HUVEC-TERT cells is mediated by CD209L/LSIGN [1]. Therefore, we also examined CD209L expression in HUVEC-TERT and PMVEC. The results showed that CD209L was expressed in HUVEC-TERT and PMVEC and that the knockdown of CALR did not interfere with the expression of CD209L levels (Figure 4A). HUVEC-TERT cells expressing CALR shRNA were infected at a higher rate than the parental cells expressing control shRNA (corresponding to two independent experiments, triplicates per group) (Figure 4B). Similarly, PMVEC expressing CALR-shRNA showed a higher rate of infection compared to their parental control cells (Figure 4C). The data suggest that expression of CALR in endothelial cells plays a rate-limiting function in SARS-CoV-2 infection.

CALR is a major calcium -binding protein whose activity plays a central role in the regulation of intracellular Ca^2+^ homeostasis [26,32]. Dysregulation of calcium homeostasis is linked to endothelial dysfunction [33], a condition that is also considered a major comorbidity factor in COVID-19 patients [34]. Therefore, we asked whether SARS-CoV-2 infection of endothelial cells modulates calcium homeostasis. To address this question, we engineered HUVEC-TERT cells expressing a Ca^2+^ sensor, yellow chameleon-Nano 15 (YC-Nano15), which is composed of an engineered calcium sensing domain of the YC2.60 gene [35]. In the inactive form, the YC-Nano15 sensor fluoresces in green, but once it is activated, it turns yellow [35]. Expression of YC-Nano15 sensor in the HUVEC-TERT is shown (Figure 5A). In the resting and non-infected cells (i.e., mock), there were no changes in the activation of YC-Nano15 in control shRNA and CALR-shRNA expressing HUVEC-TERT cells as measured by the ratio of YFP/GFP (Figure 5B). However, SARS-CoV-2 infection of HUVEC-TERT cells expressing control shRNA showed a significantly higher ratio of YFP/GFP, but cells expressing CALR shRNA under similar conditions showed markedly reduced intracellular calcium release (Figure 5B), demonstrating that CALR facilitates SARS-CoV2-induced intracellular calcium homeostasis.

## 4. Discussion

This study demonstrates that CALR interacts with the SARS-CoV-2 spike protein and exerts a negative regulatory role over spike protein levels. CALR-mediated regulation of the spike protein may represent a novel molecular mechanism by which the ER-lysosome pathway maintains the cellular proteostasis state of the spike protein. On the other hand, CALR downregulation or inactivation could ameliorate spike protein levels by modulating its lysosome-dependent degradation (Figure 6). This in turn could ultimately determine the infectivity of SARS-CoV-2 in target cells. CALR is an ER-localized calcium -binding chaperone and is well-known for its roles in protein folding and quality control. In this role, CALR recognizes and controls the quality of newly synthesized glycoproteins, ensuring that only properly folded proteins can exit from the ER to the Golgi [36,37,38].

One important and interesting aspect of our finding is that CALR uniquely recognizes the S-RBD via its P (proline-rich) domain, which is not involved in carbohydrate recognition [39]. Our homology modeling revealed that S-RBD resembles the EVH1 domain and the EVH1 lookalike fold of S-RBD, although their overall folding is not fully identical, and could account for its distinct interaction with CALR. Depleting CALR by shRNA increased the turnover of the spike protein and inhibition of lysosome-mediated proteolysis with bafilomycin-A1 in CALR depleted cells further increased spike protein levels, indicating that decoupling of the association of spike with CALR protects it from the lysosomal degradation. These data suggest that spike protein levels and turnover are closely linked to its ability to interact with CALR.

Central to SARS-CoV-2 infection is the ability of the spike protein to recognize and interact with the host entry receptors such as ACE2 and CD209L and attachment factors such as vimentin [1,2,3,4]. In this present study, we found that reducing CALR levels by shRNA improves SARS-CoV-2 infection in human umbilical and pulmonary microvascular endothelial cells, indicating that spike protein proteostasis orchestrated by CALR plays an important rate-limiting function in determining the infectivity of SARS-CoV2. Expression of CALR also regulates SARS-CoV-2-induced intracellular calcium homeostasis in endothelial cells. Knockdown of CALR in endothelial cells reduced intracellular calcium release, indicating that CALR plays a multifunctional role in SARS-CoV-2 infectivity and virus-induced host responses. It was proposed that CALR is responsible for more than 50% of the ER calcium storage [40]. Hence, ER reduced intracellular calcium, in part, could account for the decreased SARS-CoV-2-induced intracellular calcium release leading to activation of endothelial cells.

Endothelial cell activation/dysfunction has been linked to the pathology of COVID-19 such as vascular thrombosis, altered angiogenesis, and cell death [34,41,42], and intracellular calcium homeostasis plays an essential role in this cellular event [43]. Intriguingly, CALR is frequently mutated in various human cancers such as myeloproliferative neoplasms (MPNs) [44,45,46], and a recent study shows that COVID-19 patients with MPNs are significantly at higher risk of arterial thrombosis and death compared to non-MPN patients [47]. However, a link between the severity of COVID-19 and CALR mutations has not been established and requires further investigation. Additionally, the mechanisms by which CLAR may contribute to SARS-CoV2 infection require further studies. In addition to its interaction with the spike protein, CALR could also recognize other SARS-CoV2 proteins. Furthermore, CALR has been reported to be expressed extracellularly in certain cell types, particularly in cells undergoing apoptosis [28], raising the possibility that CALR could also interact with SARS-CoV2 S extracellularly. However, we did not detect the expression of CLAR in the extracellular region in endothelial cells (53). Additionally, considering the critical role of lysosome in the life cycle of viruses, it is important to investigate whether the interaction of the SARS-CoV2 S protein with CALR also affects the SARS-CoV-2 intracellular trafficking. SARS-CoV2 is known to be internalized and undergo intracellular trafficking within endosomes, which ultimately fuse with mature lysosomes, which is required for viral egress [48].

## 5. Conclusions

Cardiovascular complications are major clinical hallmarks of acute and post-acute COVID-19 [49,50,51,52]. The findings presented here provide a novel insight into spike protein proteostasis and the role of CALR in this process. However, additional structure–function analysis is required to establish the molecular details of interaction of the P domain of CARL with S-RBD. Moreover, further studies of the biological importance of spike protein interaction with CALR and the role of intracellular calcium release in the pathobiology of SARS-CoV2 could lead to the development of new antiviral therapeutics.

## Figures and Tables

**Figure 1 cells-12-02694-f001:**
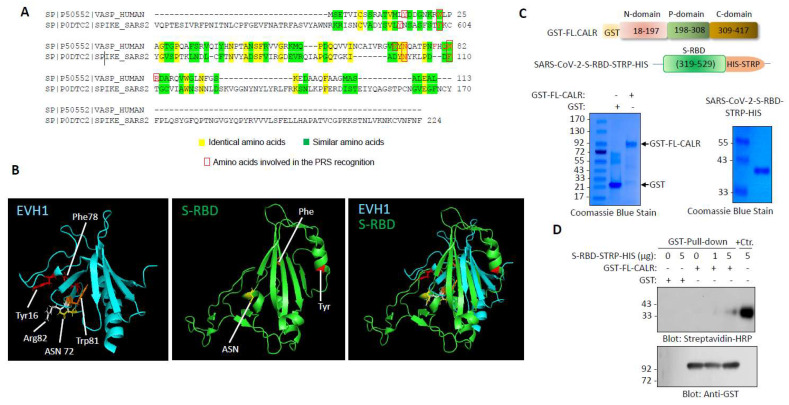
S-RBD structure resembles EVH1 domain and interacts with CALR: (**A**) Amino acid alignment of EVH1 domain of human VASP protein with SARS-CoV-2 S RBD. (**B**) Three-dimensional structures of EVH1 and S-RBD. (**C**) Schematic of GST-fusion of full length CLAR and S-RBD-STRP-HIS constructs with Coomassie blue staining of GST-CALR and S-RBD-STRP-HIS (5 µg). (**D**) GST pull-down experiment showing the binding of a purified S-RBD-STRP-HIS protein with GST-CALR.

**Figure 2 cells-12-02694-f002:**
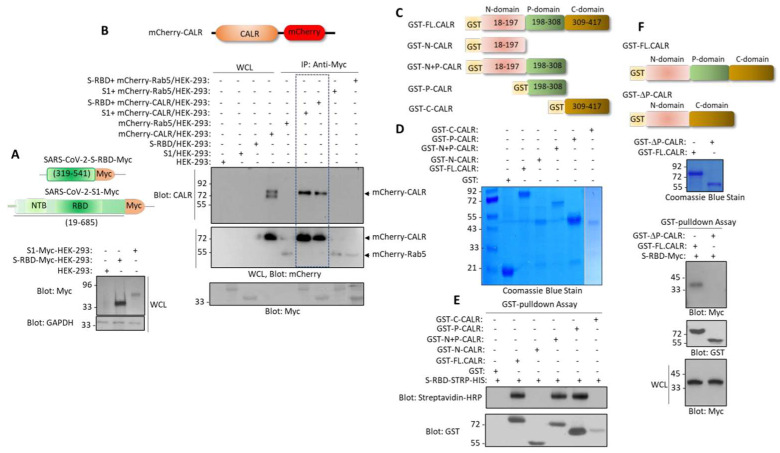
**Spike protein interacts with the P-domain of CALR.** (**A**) Schematic of S1-c-Myc subunit and S-RBD-c-Myc constructs and their expression in HEK-293 cells. (**B**) In vivo binding of S1-c-Myc and S-RBD-Myc with mCherry-CALR in HEK-293 cells. The mCherry-Rab5 construct was used as a negative control. (**C**) Schematics of GST-fusion CALR proteins. (**D**) Coomassie blue staining of GST-CALR proteins. (**E**) In vitro GST pull-down assay showing the binding of P-domain containing GST-CALR proteins with S-RBD-STRP-HIS. (**F**) Schematics of GST full-length (FL) CALR and GST-P-domain-truncated CALR (GST-ΔP-CALR), Coomassie blue staining, and the GST the pull-down assay showing that deletion of P domain abolishes the binding of CALR with S-RBD. WCL, whole cell lysate.

**Figure 3 cells-12-02694-f003:**
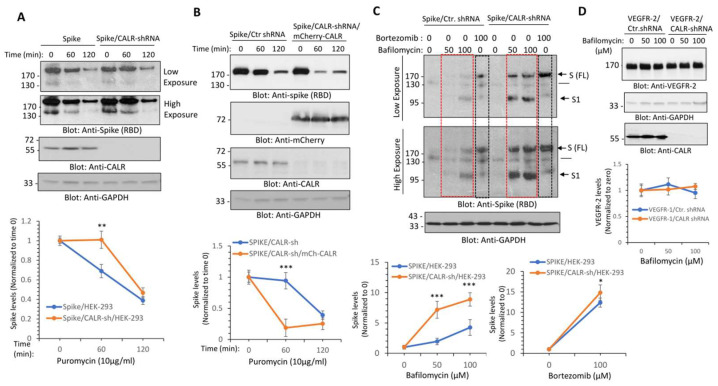
**CALR regulates the lysosome-dependent proteolysis of the spike protein.** (**A**) HEK-293 cells expressing spike-C9 alone or co-expressing spike with CALR-shRNA were treated with puromycin (10 µg/mL) for 0, 60, or 120 min. Cells were lysed and whole cell lysates were subjected to Western blot analysis using an anti-spike (RBD) or anti-CALR antibodies. Both short and long exposure of the blots are shown. The graph is a representative of three independent experiments. ** *p* < 0.01. (**B**) HEK-293 cells co-expressing spike with CALR-shRNA or spike-C9 with CALR-shRNA and mCherry-CALR were subjected to puromycin treatment and analyzed as described in panel A. (**C**) HEK-293 cells expressing spike-C9 alone or co-expressing spike with CALR-shRNA were treated with bafilomycin-1A with varying concentrations or with bortezomib (100 µM) for 12 h. Cells were lysed and whole cell lysates were subjected to Western blot analysis as shown in panel A. The graphs are representative of three independent experiments. (**D**) HEK-293 cells co-expressing VEGFR-2, control shRNA, or CALR-shRNA were treated with bafilomycin-1A with varying concentrations as indicated. Cells were lysed and whole cell lysates were subjected to Western blot analysis as shown in panel (**A**). The graph is representative of three independent experiments. Image J software (version v1.54g) was used to quantify the blots and the values of the spike protein and VEGFR-2 in each lane were normalized to the corresponding control baselines (time 0 or untreated group). * *p* < 0.05, ** *p* < 0.01 and *** *p* < 0.001.

**Figure 4 cells-12-02694-f004:**
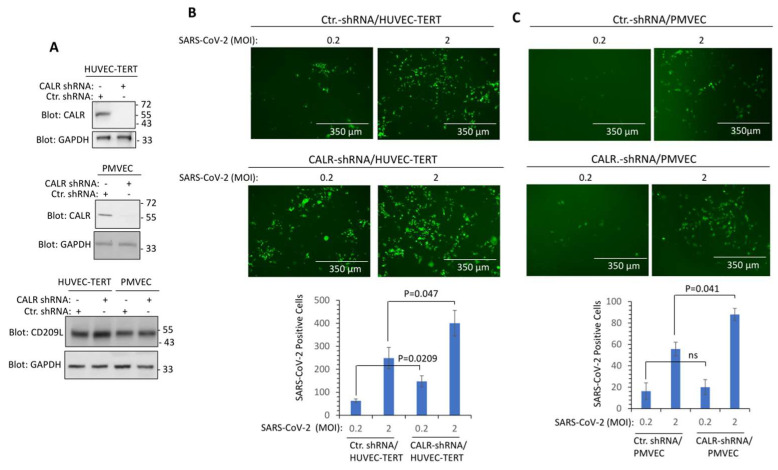
**Knockdown of CALR in human endothelial cells increases SARS-CoV-2 infection.** (**A**) Western blot analysis shows the expression of CALR in the parental and lentivirus CALR-shRNA expressing endothelial cells. (**B**) HUVEC-TERT cells expressing control shRNA or CALR-shRNA were seeded in 96-well plates (triplicate per group). The next day, the cells were infected with SARS-CoV-2-mNG at an MOI of 0.2 or 2. Twenty-four hours post-infection, cells were fixed and analyzed by fluorescence microscopy. The graph is representative of SARS-CoV-2-mNG+ cells (triplicate well per group, two independent experiments). (**C**) Human primary pulmonary microvascular endothelial cells (PMVECs) expressing control shRNA or CALR-shRNA were prepared and infected with SARS-CoV-2-mNG as shown in panel (**B**). Graph is representative of SARS-CoV-2-mNG+ cells (triplicate well per group). ns, not statistically significant.

**Figure 5 cells-12-02694-f005:**
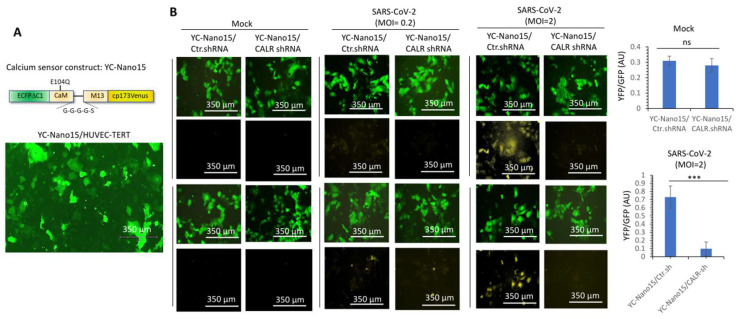
**CALR is required for SARS-CoV-2-induced intracellular calcium release in human endothelial cells.** (**A**) Expression of YC-Nano15 construct in HUVEC-TERT cells. (**B**) HUVEC-TERT cells expressing YC-Nano15 and HUVEC-TERT cells expressing CALR-shRNA with YC-Nano15 were seeded in 96-well plates (triplicate per group). The next day, the cells were infected with mock or SARS-CoV-2 at an MOI of 0.2 or 2. Twenty-four hours post-infection, the cells were fixed and analyzed by fluorescence microscopy. Two images per group are shown. GFP+ cells indicate the expression of YC-Nano15 in an inactive form. Yellow+ cells indicate the activated form of YC-Nano15. Graphs are representative of YFP/GFP ratio (triplicate well per group). ns, not statistically significant; *** *p* < 0.001.

**Figure 6 cells-12-02694-f006:**
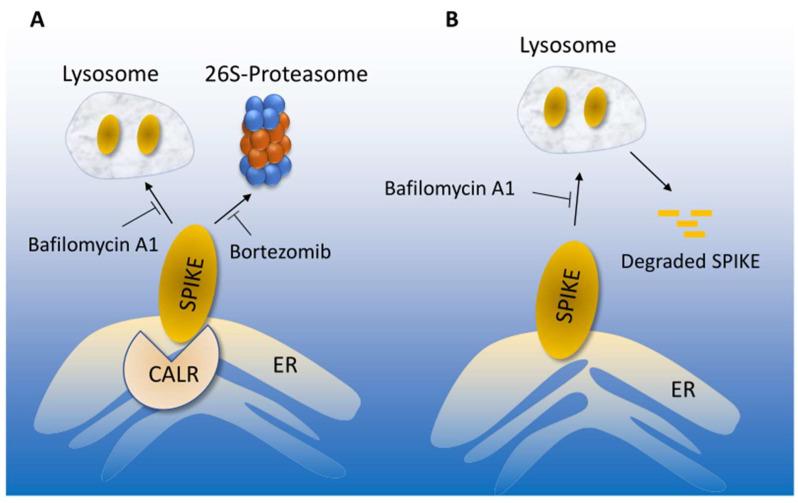
**Proposed model of CALR-mediated SARS-CoV-2 spike proteostasis.** (**A**) SARS-CoV-2 spike protein degradation is regulated by lysosome and 26S-proteosome, which could be inhibited by bafilomycin or bortezomib, respectively. Interaction of the spike with CALR mediates the lysosome-dependent proteolysis of the spike protein. (**B**) In the absence of CALR, spike protein escapes from the lysosome-mediated degradation, which could lead to increased infection.

## Data Availability

Any additional information or reagents required for the data reported in this paper are available from the corresponding author upon request.

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
