# Peer review of "Calreticulin Regulates SARS-CoV-2 Spike Protein Turnover and Modulates SARS-CoV-2 Infectivity"

_cells, 2023, doi:10.3390/cells12232694_

Round 1

Reviewer 1 Report

Comments and Suggestions for Authors

This study by Rahimi et al., demonstrate that S-RBD interacts with the proline-rich domain of CALR. The CALR regulates the S protein turnover and SARS-CoV-2 infectivity of human endothelial cells. Overall, these data are interesting, but there are some issues to be addressed before acceptance. My comments are below.

(1) Figures 1A and B are too subjective. Since there is no strong similarity, it can be deleted. The main text can also start from Line 160.

(2) In Figures 1, 2, 3, and 4, the letters and +- marks on the lanes above the gel and Western blot are misaligned and should be corrected.

(3) The inset location of Figure 4 and the legend position are misaligned. They should be corrected.

(4) What is the reason for the increased infectivity of SARS-CoV-2 in CARL-knockdown in Figure 4? Changes in expression of ACE2 and other receptor-related factors should be investigated.

(5) The interaction between CARL and spike is related to the increase in Ca ions in vascular endothelial cells during CoV-2 infection, but is this related to the prolongation of spike turnover? I don't understand what kind of relationship they have and why they are shown at the same time in this paper.

Author Response

(1) Figures 1A and B are too subjective. Since there is no strong similarity, it can be deleted. The main
text can also start from Line 160.
Thank you for the comment. However, we think it’s important to keep this figure in the manuscript.
Because this figure presents our thought process and the entire logic behind the experiments presented
in this manuscript. The key amino acids in the EVH1 domain that are involved in the recognition of the
proline rich sequences are conserved in S-RBD, albeit their folding is not fully identical. In our view
further structure-function analysis such as co-crystal structure analysis of S-RBD with CALR can provide
further molecular details about the interaction of S-RBD with CALR.
(2) In Figures 1, 2, 3, and 4, the letters and +- marks on the lanes above the gel and Western blot are
misaligned and should be corrected.
We are sorry that the figure labels were misaligned in the manuscript. It appears this occurred when the
manuscript was converted from the Microsoft word to pdf. We have corrected this issue in the current
revised manuscript. Thank you.
(3) The inset location of Figure 4 and the legend position are misaligned. They should be corrected.
Corrected. Thank you.
(4) What is the reason for the increased infectivity of SARS-CoV-2 in CARL-knockdown in Figure 4?
Changes in expression of ACE2 and other receptor-related factors should be investigated.
Our experiments suggest that CALR regulates turnover of spike protein. We think depleting CALR
increases the abondance of spike protein which could lead to robust viral replication. However, it is still
possible that CALR could contribute to increased infectivity via other mechanisms. For example, SARSCoV2 is known to get internalized and undergo intracellular trafficking within endosomes, which
ultimately fuse with mature lysosomes, which is re-quired for viral egress. Whether CALR plays a role in
SARS-CoV-2 egress is not known, which requires further studies. We have discussed this issue in the
discussion section of the manuscript. Please, note as we previously reported ACE2 is not expressed in
endothelial cells. CD209L/L-SIGN is the main entry receptor for SARS-CoV-2 in endothelial cells (Amraei
et al., ACS Central Science, 2021) . Knockdown of CALR in endothelial cells does not alter the expression
of CD209L (Figure 4A). Thank you.
(5) The interaction between CARL and spike is related to the increase in Ca ions in vascular endothelial
cells during CoV-2 infection, but is this related to the prolongation of spike turnover? I don't
understand what kind of relationship they have and why they are shown at the same time in this
paper.
Thank you for the comment. Intracellular calcium homeostasis plays a central role in endothelial cell
activation/injury. Our data as presented in this manuscript for the first time shows that infection of
endothelial cells with SARS-CoV2 leads to increased intracellular calcium release and knockdown of CALR
significantly reduced the intracellular calcium release (Figure 5B), suggesting that SARS-CoV-2-induced
intracellular calcium release requires CALR. However, from our experiments it’s not clear whether this
effect is directly linked to interaction of spike with CALR, which requires further investigation. 

Reviewer 2 Report

Comments and Suggestions for Authors

The authors worked on a very good topic and observed very interesting findings. Experiments were designed very elegantly. However, I have concerns regarding the data representation. However, authors need to improve all the figures by aligning the text carefully and increasing the font size of the text.

Author Response

Major concerns:
1. Line 219, Could authors show the proof of downregulation of CALR by immunoblot or qPCR?
Thank you for this comment. The effect of knockdown of CALR by shRNA is shown in Figure 3A via
western blot analysis.
2. Line 188-190, could the author perform the co-localization study to prove that CALR is co-localizing
with S-RBD in ER compartments?
Thank you for the suggestion. SARS-CoV-2 spike is a typical class I fusion protein and is co-translationally
translocated into the endoplasmic reticulum (Watanabe, Science et al., 220; Yamamoto et al., nt. J. Mol.
Sci. 2022) and several other groups. In our view, this particular experiment doesn’t add any new
information about the interaction of CALR, which is also a well-known ER protein.
3. During stripping methods, why blots were incubated at 50 degrees Celsius? It may degrade the
protein of interest on the blots. In my opinion, a stripping buffer is enough to strip the bound
antibodies.
This is an established method to strip/remove antibody from the membrane. Thank you.
Minor concerns:
1. In material and methods section, regarding the use of antibodies, please indicate the dilution of each
antibody to reproduce the data by other authors.
Thank you for the suggestion. We have included this information in the revised manuscript.
2. Line 318, Deregulation should be Dysregulation.
Now, it reads dysregulation. Thank you.
3. Authors need to modify all the figures in order to align the text, particularly in immunoblots where
authors have mentioned the +, - or time 0, 60 120 min. Some texts are inside the figures. In Fig. 6 model
in right side image, one arrow is inside the Bafilomycin A1.
We are deeply sorry for the quality of the images and figure labels, which appears caused during
conversion of the manuscript to pdf. We corrected all the figures and figure labels. Thank you. 

Round 2

Reviewer 2 Report

Comments and Suggestions for Authors

Authors have addressed all the issues raised by reviewers.